# Development of a Lightweight Crop Disease Image Identification Model Based on Attentional Feature Fusion

**DOI:** 10.3390/s22155550

**Published:** 2022-07-25

**Authors:** Zekai Cheng, Meifang Liu, Rong Qian, Rongqing Huang, Wei Dong

**Affiliations:** 1School of Computer Science and Technology, Anhui University of Technology, Ma’anshan 243032, China; ahut-my@ahut.edu.cn (M.L.); tty0013@ahut.edu.cn (R.H.); 2Institute of Agricultural Economy and Information, Anhui Academy of Agricultural Sciences, Hefei 230031, China; qr930@126.com (R.Q.); dw06@163.com (W.D.)

**Keywords:** crop diseases identification, ResNet18, DSGIResNet_AFF, lightweight residual blocks, inverted residual blocks, attentional feature fusion

## Abstract

Crop diseases are one of the important factors affecting crop yield and quality and are also an important research target in the field of agriculture. In order to quickly and accurately identify crop diseases, help farmers to control crop diseases in time, and reduce crop losses. Inspired by the application of convolutional neural networks in image identification, we propose a lightweight crop disease image identification model based on attentional feature fusion named DSGIResNet_AFF, which introduces self-built lightweight residual blocks, inverted residuals blocks, and attentional feature fusion modules on the basis of ResNet18. We apply the model to the identification of rice and corn diseases, and the results show the effectiveness of the model on the real dataset. Additionally, the model is compared with other convolutional neural networks (AlexNet, VGG16, ShuffleNetV2, MobileNetV2, MobileNetV3-Small and MobileNetV3-Large), and the experimental results show that the accuracy, sensitivity, F1-score, AUC of the proposed model DSGIResNet_AFF are 98.30%, 98.23%, 98.24%, 99.97%, respectively, which are better than other network models, while the complexity of the model is significantly reduced (compared with the basic model ResNet18, the number of parameters is reduced by 94.10%, and the floating point of operations(FLOPs) is reduced by 86.13%). The network model DSGIResNet_AFF can be applied to mobile devices and become a useful tool for identifying crop diseases.

## 1. Introduction

In China, agriculture is the primary industry, which is the basis of national economic construction and development and provides material security for the people. In recent years, global warming and environmental pollution have hurt the environment for crops to survive, and crop diseases have become more common and frequent [1]. If the diseases cannot be accurately and quickly identified and effective control methods are put in place, the yield and quality of crops will be greatly reduced [2].

The traditional identification of crop diseases mainly relies on manual work, and it is judged by the experience accumulated by farmers in the farming process. This method is usually time-consuming and labor-intensive and has strong subjectivity, poor real-time performance, and a high misclassification rate [3]. In order to overcome the problems of manual identification of crop diseases, with the development of machine learning, researchers first introduced machine learning into crop disease identification and used the support vector machine (SVM) [4], K-nearest neighbor (KNN) [5], random forest [6], and other methods for crop disease identification, but their feature extraction capabilities are limited. However, crop disease images usually have complex backgrounds and different features such as texture, shape, and color of disease spots, which greatly affect the effectiveness of the identification. In recent years, with the development of deep learning technology, various convolutional neural network models have been used for crop disease identification, which does not rely on specific features and can quickly extract different disease features in complex environments, providing effective methods to identify crop diseases accurately and quickly [7,8,9,10]. The crop disease identification methods based on the convolutional neural network have the advantage of high identification accuracy, but they also have the problem of a large number of parameters and high FLOPs, which makes them unable to be applied in real environments. Therefore, more and more scholars are committed to the study of small and efficient lightweight convolutional neural networks, to reduce the number of parameters and FLOPs, and improve the operational efficiency of the models [11,12,13,14].

In order to increase the application value of the crop disease identification models based on a convolutional neural network in the actual farming environment, we propose a lightweight crop disease image identification model based on attentional feature fusion DSGIResNet_AFF by optimizing the original network model ResNet18. The main contributions of this paper are as follows:
In order to satisfy the identification of rice and corn diseases in real environments, we collect four kinds of rice diseases and four kinds of corn diseases in real environments, construct a crop disease image dataset, and expand the dataset through data augmentation to improve the generalization ability of the model in practical applications.In this paper, we propose a lightweight convolutional neural network model for identifying crop diseases. The model introduces self-built lightweight residual blocks, inverted residual blocks, and attentional feature fusion modules, which greatly improve the feature extraction ability of the model. Compared with the original ResNet18 model, it not only greatly reduces the number of parameters and FLOPs of the model, but also improves the ability of the model to identify crop diseases.To prove the effectiveness of the proposed model, it is compared with various other classical network models (AlexNet [15], VGG16 [16], ShuffleNetV2 [17], MobileNet-V2 [18], MobileNetV3-Small and MobileNetV3-Large [19]), and the experimental results show that the overall effect of the proposed network model DSGIResNet_AFF is optimal.


The network model DSGIResNet_AFF can be applied to the actual farming environment to help farmers identify crop diseases quickly and accurately, provide effective technical means to make disease control strategies scientifically and improve crop yield and quality, and play a positive role in promoting smart agriculture, with practical application value.

## 2. Related Work

With the development of deep learning technology quickly, convolutional neural networks have been used in crop disease identification widely due to their powerful feature extraction capabilities. Brahimi, M. et al. [20] used the pre-trained models of AlexNet and GoogleNet to identify tomato leaf diseases, and the accuracy reaches 99.18%. Chen, J. et al. [21] proposed an improved VGG model (INC-VGGN) based on the VGG model by introducing the inception module, adding pooling layers, and modifying the activation function, and its accuracy of rice disease prediction in the complex background is 92%. Nithish Kannan, E. et al. [22] used data augmentation to expand the tomato disease dataset and introduced transfer learning into the training process, and used the fine-tuned ResNet50 model to identify the diseases on the leaves in the dataset, and the accuracy reaches 97%. Zhao, S. et al. [23] proposed the SE-ResNet50 model, which was based on the ResNet50 network model with the addition of the SE module, and the accuracy of the model for identifying tomato leaf diseases reaches 96.81%. ZHU, S. et al. [24] introduced a multi-scale convolution module and SE attention module on the basis of the residual network, and the average accuracy reaches 99.4% when identifying fruit tree leaves diseases.

Although the above methods have achieved high identification accuracy, the improved identification models have a complex structure, high complexity, and high training overhead, and cannot be applied in the actual environments. In recent years, in order to balance the accuracy and complexity of the convolutional neural network models, small and efficient lightweight convolutional neural networks have gradually emerged and become an active research field. Rahman, C.R. et al. [25] proposed a two-stage small CNN architecture and compared it with state-of-the-art memory-efficient CNNs. Additionally, the accuracy of the proposed architecture achieves 93.3% while significantly reducing the model size. Chen, J. et al. [26] proposed a new network model Mobile-DANet based on DenseNet, which retained the structure of transition layers and used depthwise separable convolutions in dense blocks instead of traditional convolutions layer and embedded the attention module. The experimental results show that the model has an average accuracy of 95.86% in identifying corn crop disease images under complex backgrounds. Bao, W. et al. [27] constructed a convolution neural network model SimpleNet using convolution and the inverted residual blocks, introduced the CBAM module into the inverted residual blocks, and proposed a feature fusion module to reduce the loss of the detailed disease features. The experimental results show that the proposed model achieves an identification accuracy of 94.1% on the wheat ear disease dataset with only 2.13 M parameters. JIA, H. et al. [28] introduced depthwise separable convolution and global average pooling in the VGG network. The improved network has higher disease recognition accuracy on the PantVillage dataset while reducing the model complexity. Zeng, W. et al. [29] proposed a lightweight dense scale network model (LDSNet) that can be used to identify corn diseases in complex backgrounds, and the core module of the model was the improved dense dilated convolution (IDDC), and a new loss function was proposed to optimize the network model, the accuracy of the model reaches 95.4%, and the number of parameters only accounts for 45.4% of ShuffleNetV2. Although the above studies have achieved good results, there is still room for improvement in the models.

## 3. Materials and Methods

In this section, we introduce the dataset and describe the design and development of the model used for crop disease identification.

### 3.1. Dataset and Preprocessing

The experimental dataset is mainly from a provincial agricultural research institute, and some corn disease images come from the public PlantVillage dataset to form a usable dataset. The dataset has a total of 6200 images, including two crops of rice and corn, with 8 disease categories. The disease categories are Rice Brown Spot (10000), Rice Leaf Sheath Rot (10001), Rice False Smut (10002), Rice Blast (10003), Corn Gray Leaf Spot (10004), Corn Curvularia Leaf Spot (10005), Corn Common Rust (10006) and Corn Northern Leaf Blight (10007).

In order to avoid overfitting of the model and improve the generalization ability of the model, the data augmentation techniques are performed on some disease images to expand the sample data, and the data augmentation operations of the original dataset are including flipping, random rotation, random adjustment of brightness and contrast, random adding salt and pepper noise (SP noise), random erasing. The expanded dataset has 7940 images, and the dataset distribution is shown in Table 1. Figure 1 shows the original samples and data augmented images. During the experiment, the images are uniformly scaled to 224 × 224 pixels, and the dataset is divided into a training set and a test set at the ratio of 4:1.

### 3.2. ResNet18

In order to solve a series of problems such as the problem of vanishing/exploding gradients caused by the increase in network depth, He, K. et al. [30] proposed the residual network in 2015, which successfully solved these problems. The core structure of the residual network is the residual block, as Figure 2, which connects the input feature X with the F(X) obtained from the stacked weight layers across the layers to obtain the output H(X)=F(X)+X.

### 3.3. Lightweight Residual Blocks

#### 3.3.1. Depthwise Separable Convolution

The depthwise separable convolution [31] is the module that decomposes the standard convolution into depthwise convolution and 1×1 convolution called pointwise convolution. The standard convolution can extract and combine features at one time to obtain new outputs. However, the depthwise separable convolution divides this into two steps. One step is to convolve a single channel, respectively, using the same number of filters as input channels, and the other step uses 1×1 convolution to combine the output of features from the first step to obtain new outputs.

The convolution process of standard convolution is shown in Figure 3. Standard convolution takes feature maps of size $D_{F}\times D_{F}\times M$ as input, and then extracts feature information using filters to obtain $D_{F}\times D_{F}\times N$ feature maps as output. M and N are the number of input channels and output channels, respectively, and the size of the filter is $D_{K}\times D_{K}\times M$. The number of parameters and FLOPs of standard convolution are calculated as follows:(1)$$Params_{SC}=D_{K}\cdot D_{K}\cdot M\cdot N=D_{K}{}^{2}MN$$
(2)$$FLOPs_{SC}=D_{K}\cdot D_{K}\cdot M\cdot N\cdot D_{F}\cdot D_{F}=D_{K}{}^{2}MND_{F}{}^{2}$$

The convolution process of depthwise separable convolution is shown in Figure 4. The size of the input feature maps is $D_{F}\times D_{F}\times M$, and the output feature maps with the size $D_{F}\times D_{F}\times N$ are obtained through depthwise convolution and pointwise convolution. The number of parameters and FLOPs of depthwise separable convolution are calculated as follows:(3)$$Params_{DSC}=D_{k}\cdot D_{k}\cdot M+1\cdot 1\cdot M\cdot N=D_{K}{}^{2}M+MN$$
(4)$$FLOPs_{DSC}=D_{k}\cdot D_{k}\cdot M\cdot D_{F}\cdot D_{F}+1\cdot 1\cdot M\cdot N\cdot D_{F}\cdot D_{F}\cdot D_{F}=D_{K}{}^{2}MD_{F}{}^{2}+MND_{F}{}^{2}$$

In summary, the ratio of the number of parameters and FLOPs of the depthwise separable convolution to the standard convolution can be expressed as:(5)$$\frac{Params_{DSC}}{Params_{SC}}=\frac{D_{K}{}^{2}M+MN}{D_{K}{}^{2}MN}=\frac{1}{N}+\frac{1}{D_{k}{}^{2}}$$
(6)$$\frac{FLOPs_{DSC}}{FLOPs_{SC}}=\frac{D_{K}{}^{2}MD_{F}{}^{2}+MND_{F}{}^{2}}{D_{K}{}^{2}MND_{F}{}^{2}}=\frac{1}{N}+\frac{1}{D_{k}{}^{2}}$$

From Equations (5) and (6), we can see that if the size of convolution kernel is 3 × 3, the number of parameters and FLOPs of the depthwise separable convolution are about 1/9 of the standard convolution. With a slight loss of accuracy, the complexity of the model is reduced, and making the model more efficient.

#### 3.3.2. Group Convolution

The group convolution [32] is to group the input multi-channel feature maps, then use the grouped filters to convolve the corresponding groups, and finally combine the convolution results as the output feature maps. The process of group convolution is shown in Figure 5, where g represents the number of groups. Compared with the standard convolution (Section 3.3.1), the size of filters becomes $D_{K}\times D_{K}\times M/g$. The number of parameters and FLOPs are calculated as follows:(7)$$Params_{GC}=D_{K}\cdot D_{K}\cdot M/g\cdot N/g\cdot g=1/g\;Params_{SC}$$
(8)$$FLOPs_{GC}=D_{K}\cdot D_{K}\cdot M/g\cdot N/g\cdot D_{F}\cdot D_{F}\cdot g=1/g\;FLOPs_{SC}$$

The group convolution reduces the number of parameters and FLOPs to 1/g of the standard convolution, which not only reduces the complexity of the model, but also improves the identification accuracy of the model. Through group convolution, feature information with different focus points can be obtained, which can express the features of the input images more completely. At the same time, in the group convolution, the filters with high correlation are grouped into a block diagonal structure, which avoids the occurrence of overfitting, similar to regularization, so that the optimizer can learn a more accurate and efficient network. 

#### 3.3.3. The Structure of Lightweight Residual Block

The lightweight residual (DSGRes) block shown in Figure 6 is composed of depthwise separable convolution and group convolution. The introduction of this module drastically reduces the number of parameters and FLOPs of the model. Additionally, the group convolution compensates for the accuracy loss caused by depthwise separable convolution. In crop disease images, the area, color, shape and other characteristics of disease spots are different. Through this module, the feature information of different points of interest can be obtained, and various disease spot feature information can be extracted, thereby improving the disease identification ability of the model. 

### 3.4. Inverted Residual Blocks

In convolutional neural networks, adjusting the structure of network structure to obtain a balance between model accuracy and performance has been an important research area in the past few years. Sandler, M. et al. [18] proposed the inverted residual structure, which not only reduces the complexity of the network, but also improves the accuracy of the model. This module expands the input low-dimensional channel features, then uses depthwise convolution to convolve the expanded features, and finally projects the high-dimensional features back into the low-dimensional space.

The structure of the inverted residual blocks is shown in Figure 7, which differs from the traditional residual blocks in the following ways:
The inverted residual block uses 3×3 depthwise convolution instead of the traditional 3×3 convolution to reduce the parameters of the model.In the first two convolutional layers of the inverted residual block, ReLU6 is used to replace the ReLU activation function in the traditional convolution. ReLU6 has stronger robustness under low precision computation, and its expression is as follows:(9)ReLU6=min(max(0,x),6)In the linear bottleneck structure of the inverted residual block, 1×1 pointwise convolution is used to map high-dimensional feature information into low-dimensional space. When outputting low-dimensional features, if the ReLU activation function is used, it is easy to cause information loss. Therefore, a linear activation function is selected to replace the ReLU to avoid the loss of feature information.


The detailed operation parameters of each layer of the inverted residual block are shown in Table 2, where h and w represent the height and width of the input feature map, C and C′ represent the input and output channels, respectively, S represents the step size, and t represents the expansion factor, which is used to expand the number of channels.

### 3.5. Attentional Feature Fusion

Feature fusion is the combination of features from different network layers or different branches, and is a ubiquitous part of modern network architectures, usually achieved by addition or concatenation operations. Feature fusion in most attention modules is achieved by combining features in different convolution kernels, groupings, or network layers, only the feature maps are fused fixedly without considering cross-layer feature fusion, and the fusion weights in the model are produced by the global channel attention, which will weaken the fine feature information of the target area, and thus cannot improve the identification effect of network model [33,34].

The attentional feature fusion (AFF) [35] effectively solves the above problem. As shown in Figure 8, the AFF module adds a branch to extract local features based on global channel attention and completes the fusion of local features and global features in the attention module. The module uses pointwise convolution to obtain multi-scale feature contexts, which can not only retain and highlight fine details in low-level features, but also save the number of parameters as much as possible, keep lightweight, and can be used to replace existing feature fusion modules in network models such as InceptionNet, ResNet, FPN and other network models. Attentional feature fusion provides a general and effective method to improve the performance of the network model. The global channel context $G\left(X\right)\in R^{H\times W\times C}$, and the local channel context $L\left(X\right)\in R^{H\times W\times C}$ are calculated as follows:(10)$$G\left(X\right)=B\left(PWConv_{2}\left(\delta \left(B\left(PWConv_{1}\left(GAP\left(X\right)\right)\right)\right)\right)\right)\;$$
(11)$$L\left(X\right)=B\left(PWConv_{2}\left(\delta \left(B\left(PWConv_{1}\left(X\right)\right)\right)\right)\right)\;\;$$
where GAP stands for GlobalAvgPooling, *δ* is ReLU activation function. $PWConv_{1}$ and $PWConv_{2}$ represent PW convolution, and the filters sizes are H×W×C/r and H×W×C, respectively, (when extracting global channel context information, H=W=1).

For given two feature maps $X,Y\in R^{H\times W\times C}$, attentional feature fusion can be expressed as:(12)$$\begin{array}{l} Z=M\left(X⊕Y\right)⊗X+\left(1-M\left(X⊕Y\right)\right)⊗Y\\ \;\;\;\;=\sigma \left(G\left(X⊕Y\right)⊕L\left(X⊕Y\right)\right)⊗X+\left(1-\sigma \left(G\left(X⊕Y\right)⊕L\left(X⊕Y\right)\right)\right)⊗Y \end{array}$$
where $Z\in R^{C\times H\times W}$ is the output after weighted feature fusion, σ is Sigmoid activation function, ⊕ represents the broadcasting addition and ⊗ represents the element-wise multiplication, M(X⊕Y)∈(0,1) is the fused channel weights generated by AFF, which are used to weight the input feature maps.

We use the AFF module to replace the feature fusion module in the original residual block. The structure is shown in Figure 9. In the AFF module, X is the output of the identity mapping in the residual block, and Y is the output of the learned residual part in the residual block. The introduction of the AFF module makes the model focus on the global features and local features of the input images, extracts richer and finer feature information, suppresses other irrelevant noise in the input image, locates the target area quickly and accurately, and improves the disease identification ability of the model.

### 3.6. Overall Model Structure

The DSGIResNet_AFF network model designed in this paper is based on the ResNet18 model and introduces DSGRes blocks, inverted residual blocks, and AFF modules. The final model structure is shown in Figure 10. The residual block1 and residual block2 of the original ResNet18 model are replaced with inverted residual blocks, residual block3 and residual block4 are replaced with DSGRes blocks, and the AFF module is used to replace the feature fusion module of all residual blocks.

## 4. Results and Discussion

### 4.1. Ablation Experiment

In this paper, we use ResNet18 as the basic network model, use various methods to improve it and use the improved network model DSGIResNet_AFF to identify various crop disease images. In order to prove the advantages of the improved model, several sets of comparative experiments were conducted, and the results are shown in Table 3.

It can be observed from Table 3 that the ResNet18_1/2 model can achieve the purpose of reducing the number of parameters and FLOPs by reducing the number of channels of the basic network model ResNet18. However, with the reduction in the number of channels, the feature extraction ability of the model becomes worse, and the useful crop disease information that can be extracted will be reduced, so its accuracy, sensitivity, F1-score, and AUC are reduced compared to the ResNet18. DSGResNet uses the lightweight residual blocks (DSGRes blocks) to replace the residual blocks of ResNet18, which further reduces the number of parameters and FLOPs of the model while maintaining the same accuracy, sensitivity, F1-score and AUC as ResNet18_1/2. DSGIResNet introduces the inverted residual blocks on the basis of the DSGResNet, and the accuracy, sensitivity, F1-score and AUC of the model are improved while the number of parameters and the FLOPs are reduced, which solves the problem of worse feature extraction ability of the model caused by the reduction in the number of channels of the network model. Finally, the DSGIResNet_AFF model constructed in this paper, although its number of parameters and FLOPs have a slight increase compared with the DSGIResNet model, compared with the basic network model ResNet18, its accuracy, sensitivity, F1-score and AUC have increased by 0.63%, 0.57%, 0.6%, and 0.03%, respectively, and its number of parameters is reduced by 94.10%, and the FLOPs is reduced by 86.13%.

The use of the DSGRes blocks and the inverted residual blocks enables the model to focus on more abundant disease features, reducing the number of parameters and FLOPs while enhancing the feature extraction capability of the model. The introduction of the AFF module enables the model to extract various fine disease spot features, locate the target area accurately and quickly, and suppress the noise in the images that would cause interference to the network model, which can improve the ability of the model to identify crop diseases. As shown in Figure 11, the area concerned by the model after adding the AFF module overlaps with the disease spot region, and accurately locates the target area of the images, while the basic network model ResNet18 has a relatively weak ability to locate the disease spot region, and sometimes will incorrectly locate the center of the target area.

### 4.2. Comparison Experiment of Different Identification Models

To objectively evaluate the capabilities of the proposed network model in this paper, its experimental results are compared with the network models of AlexNet, Shufflenet V2, VGG16, MobileNetV2, MobileNetV3-Small and MobileNetV3-Large. The experimental results of the different models are shown in Table 4.

It can be observed from Table 4 that although the FLOPs of the crop disease identification model DSGIResNet_AFF proposed in this paper are slightly higher than that of MobileNetV3-Small and MobileNetV3-Large, their number of parameters are the smallest, and its accuracy, sensitivity, F1-score, and AUC performance are optimal. From the comprehensive overall evaluation criteria, it can be obtained that the crop diseases identification model DSGIResNet_AFF proposed in this paper has a much lower number of parameters and FLOPs than other identification models and has higher accuracy, sensitivity, F1-score, and AUC. In this paper, by improving the traditional ResNet18 model, the number of parameters and FLOPs of the model is reduced, and the feature extraction ability of the model for different disease spots is enhanced, so that the model can focus on disease spots of different scales and shapes and avoid the appearance of overfitting in the training process, thus improving the overall performance of the model. 

## 5. Conclusions

To further improve the efficiency and accuracy of crop disease identification, this paper proposes a lightweight crop disease image identification model based on attentional feature fusion DSGIRseNet_AFF. Based on the ResNet18 model, the model uses the DSGRes blocks and the inverted residual blocks to replace the residual blocks in the backbone network, which improves the feature extraction ability of the model while reducing the number of parameters and FLOPs of the model and then uses the AFF module to replace the feature fusion module in the basic model, which enables the model to focus on richer and finer crop disease features, quickly locate the target area, and improve the identification ability of the model. In the identification of rice and corn disease images, the convolutional neural network model DSGIResNet_AFF proposed in this paper is better than other well-known image identification models, with an accuracy of 98.3%, a sensitivity of 98.23%, F1-score of 98.24%, and the AUC of 99.97%. At the same time, the number of parameters and FLOPs are greatly reduced compared to the basic network model ResNet18, which greatly saves computational resources and is more suitable for deployment on mobile devices to diagnose crop diseases. In the real agricultural environment, it can help farmers to identify the types of crop diseases quickly and accurately, so as to formulate corresponding control measures and reduce the loss of crops, which has practical application value.

## Figures and Tables

**Figure 1 sensors-22-05550-f001:**
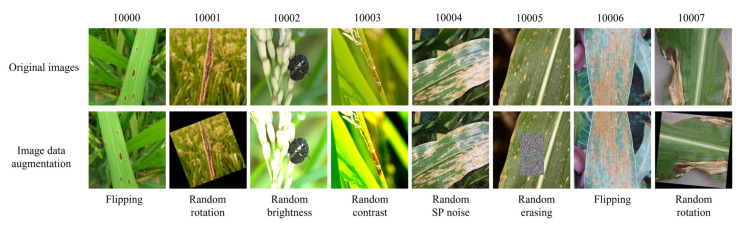
Data augmentation of different crop disease images.

**Figure 2 sensors-22-05550-f002:**
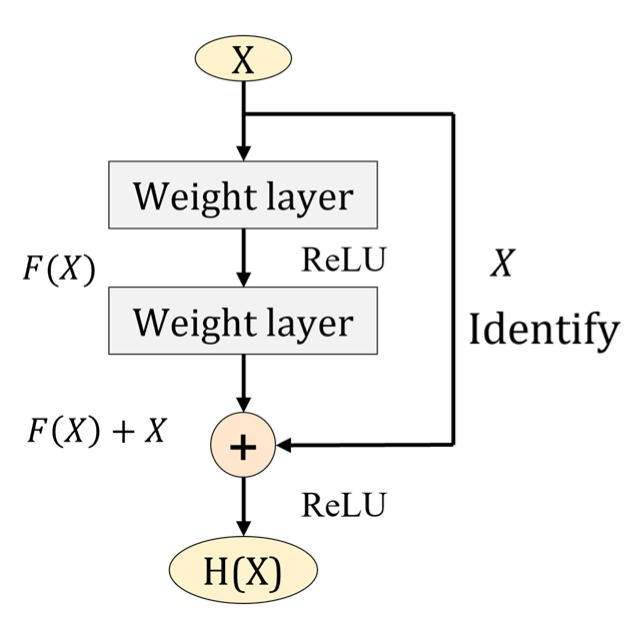
Residual block.

**Figure 3 sensors-22-05550-f003:**
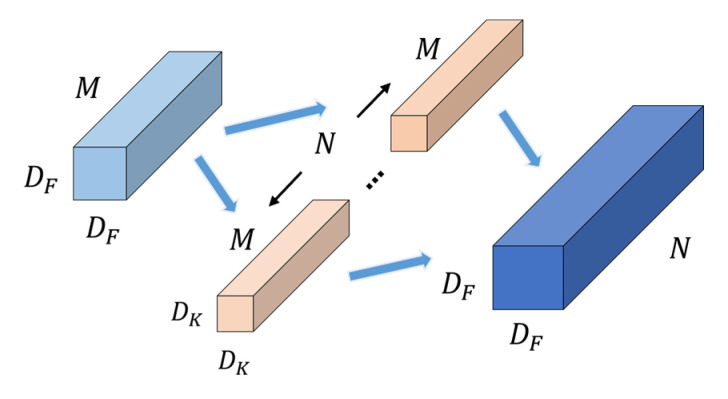
Standard convolution. The feature information of the input feature map is extracted using $D_{k}\times D_{k}\times M$ filters, and the output feature map $D_{F}\times D_{F}\times N$ is obtained, where M and N are the number of channels of the input and output feature maps, respectively.

**Figure 4 sensors-22-05550-f004:**
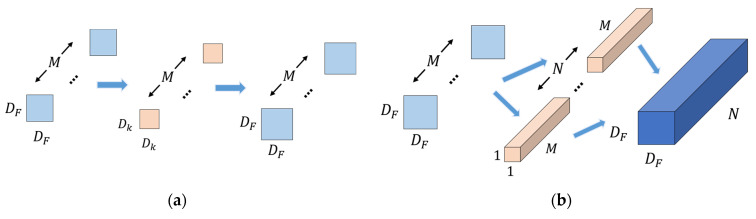
Depthwise separable convolution. (**a**) Depthwise convolution. Convolve the single-channel by using $D_{k}\times D_{k}\times 1$ filters with the same number of input channels to obtain the single-channel feature map $D_{F}\times D_{F}\times M$. (**b**) Pointwise convolution. Take the single channel feature map obtained by the depthwise convolution as input and use 1×1×M filters to combine them to obtain the output feature map $D_{F}\times D_{F}\times N$.

**Figure 5 sensors-22-05550-f005:**
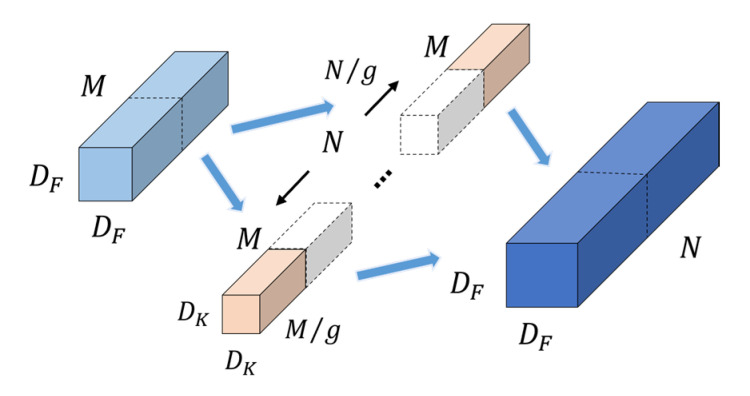
Group convolution. The *g* independent groups of N/g filters operate on part of the channels M/g of the corresponding input feature map, changing the size of the filter from $D_{k}\times D_{k}\times M$ to $D_{k}\times D_{k}\times M/g$ (*g* stands for the number of groups). This change does not affect the size of the input and output feature maps, but greatly reduces the complexity of the model.

**Figure 6 sensors-22-05550-f006:**
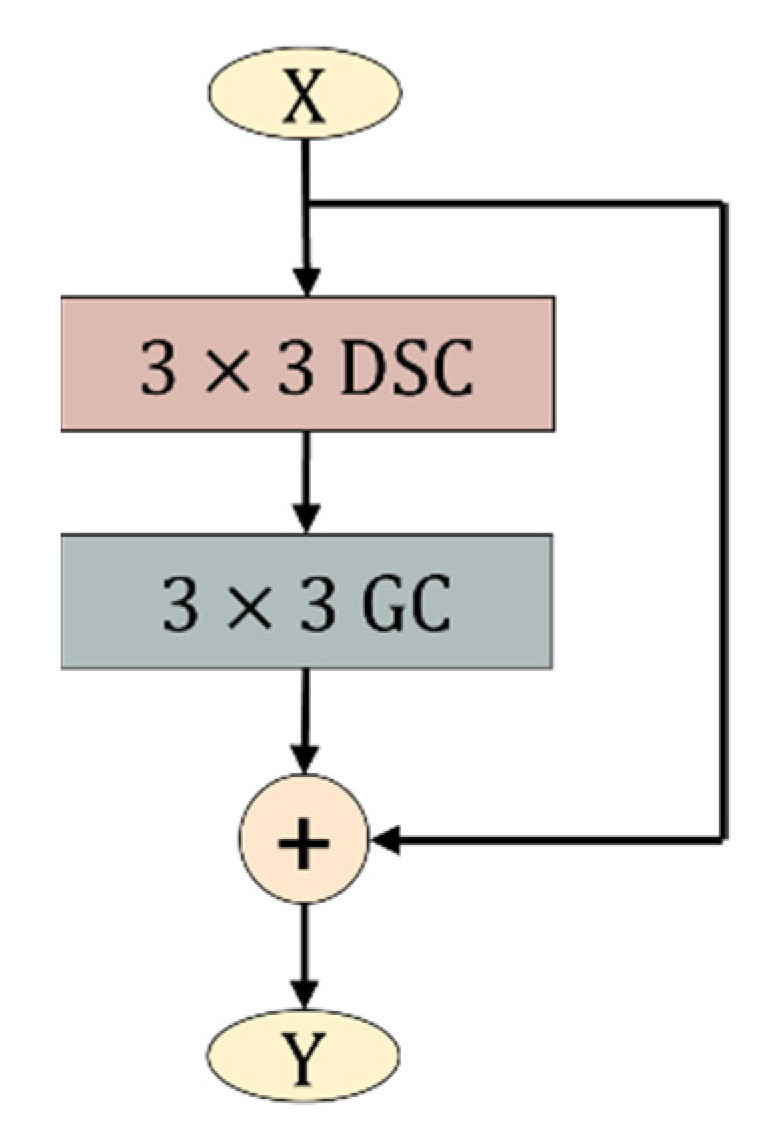
Lightweight residual (DSGRes) block. The DSC stands for the depthwise separable convolution, and the GC stands for the group convolution.

**Figure 7 sensors-22-05550-f007:**
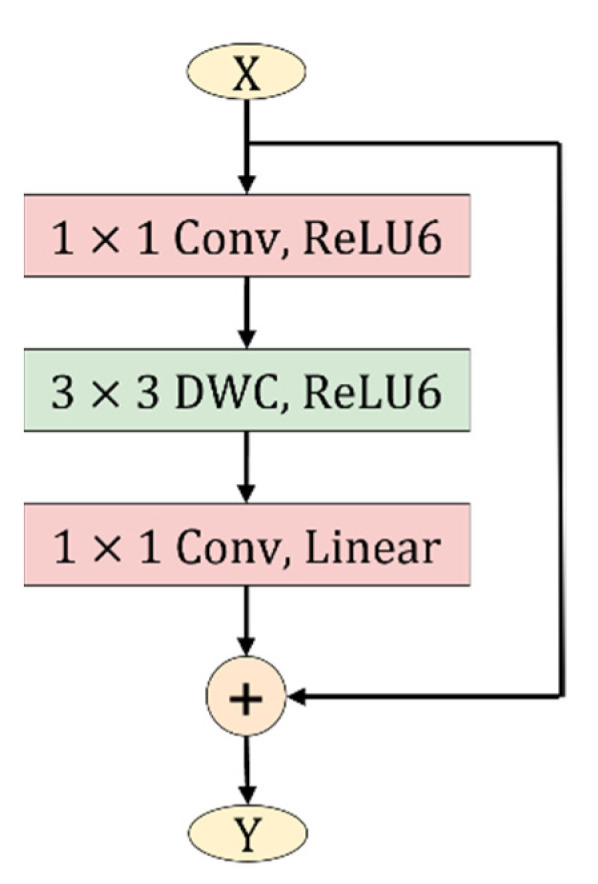
Inverted residuals block.

**Figure 8 sensors-22-05550-f008:**
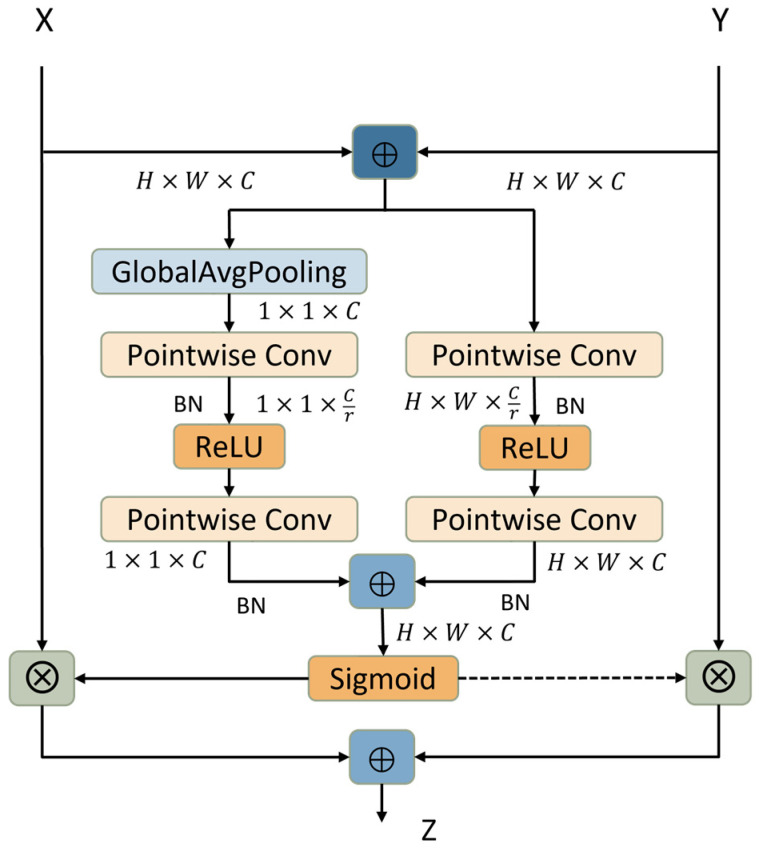
Attentional feature fusion module. First, perform initial feature fusion on the input features X and Y, and input the results into the left and right branches to extract global context information and local context information, respectively; then, fuse the information extracted from the two branches and input them into the Sigmoid activation function to obtain fusion channel weights; finally, use the fusion channel weights to weight the original input features to obtain the weighted feature maps of X and Y, respectively, and then add the results along the channel to obtain the final feature information.

**Figure 9 sensors-22-05550-f009:**
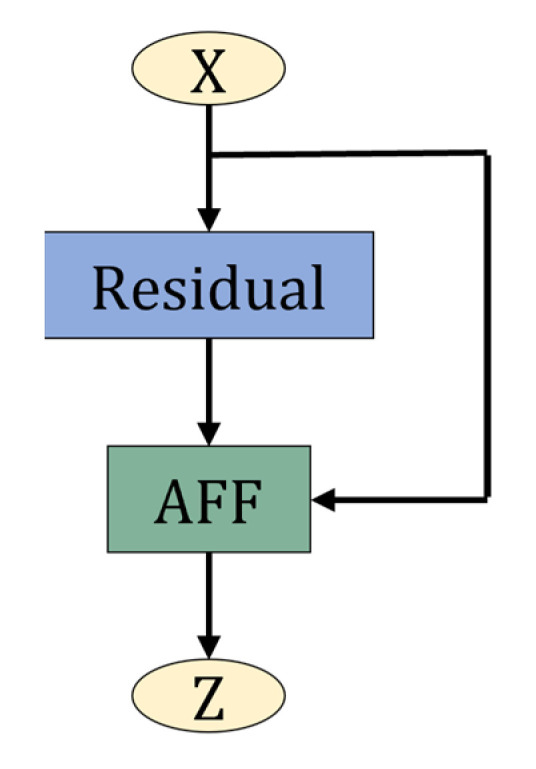
AFF residual block. The feature fusion module in the original residual block is replaced with the attentional feature fusion (AFF) module to improve the model’s ability to localize the target.

**Figure 10 sensors-22-05550-f010:**
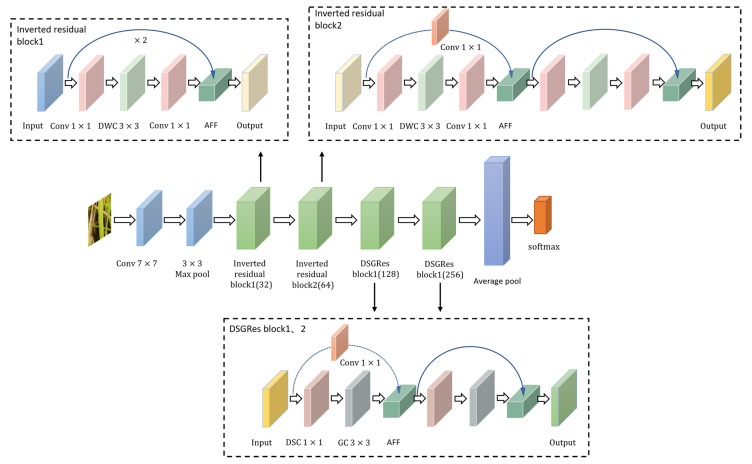
DSGIResNet_AFF network model structure. The core of the model is the two inverted residual blocks (Section 3.4) and the two lightweight residual (DSGes) blocks (Section 3.3). DWC stands for the depthwise convolution and DSC stands for the depthwise separable convolution (Section 3.3.1), GC stands for the group convolution (Section 3.3.2), and AFF stands for the attentional feature fusion (Section 3.5).

**Figure 11 sensors-22-05550-f011:**
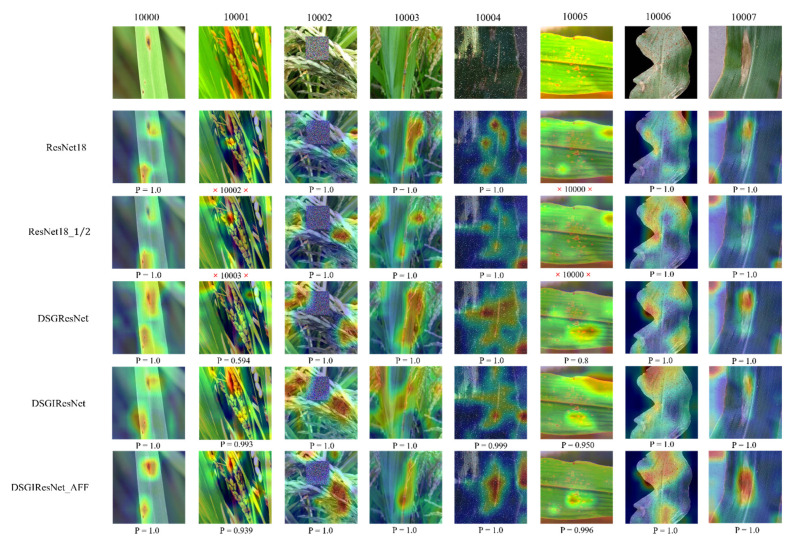
The heatmap shows the areas of interest of the model under different improvement methods. The incorrectly predicted images are indicated by the symbol **×**. The class name of the incorrect prediction and the correctly predicted softmax scores (P) are shown at the bottom of the heat map.

**Table 1 sensors-22-05550-t001:** Dataset distribution.

Classes (Disease)	Numbers of Original Dataset	Number of Expend Dataset
Rice Brown Spot (10000)	1100	1100
Rice Leaf Sheath Rot (10001)	160	1100
Rice False Smut (10002)	940	940
Rice Blast (10003)	900	900
Corn Gray Leaf Spot (10004)	500	1000
Corn Curvularia Leaf Spots (10005)	700	1000
Corn Rust (10006)	1000	1000
Corn Northern Leaf Blight (10007)	900	900
Total	6200	7940

**Table 2 sensors-22-05550-t002:** The per-layer convolution parameters of inverted residual blocks.

Input	Operation	Output
h×w×C	1×1 Conv, ReLU6	h×w×tC
h×w×tC	3×3 DW Conv s = s, ReLU6	h×w×tC
$\frac{h}{s}\times \frac{w}{s}\times tC$	1×1 Conv, Linear	$\frac{h}{s}\times \frac{w}{s}\times C^{\prime }$

**Table 3 sensors-22-05550-t003:** Comparison of experimental results of different improved methods. Accuracy, sensitivity, F1-score, AUC, Params and FLOPs are the evaluation criteria of the network model, and channels represent the number of channels taken by each residual block of the model.

Models	Accuracy (%)	Sensitivity (%)	F1-Score (%)	AUC (%)	Params (M)	Flops (G)	Channels
ResNet18	97.67	97.66	97.64	99.94	11.148	1.694	[64,128,256,256]
ResNet18_1/2	97.54	97.48	97.50	99.93	2.930	0.541	[32,64,128,256]
DSGResNet	97.54	97.51	97.50	99.93	0.542	0.280	[32,64,128,256]
DSGIResNet	97.98	97.90	97.92	99.95	0.485	0.222	[32,64,128,256]
DSGIResNet_AFF	98.30	98.23	98.24	99.97	0.658	0.235	[32,64,128,256]

**Table 4 sensors-22-05550-t004:** Experimental comparison of different network models on crop disease datasets.

Models	Accuracy (%)	Sensitivity (%)	F1-Score (%)	AUC (%)	Params (M)	Flops (G)
AlexNet	94.65	94.51	94.55	99.63	58.270	0.667
Shufflenet V2	95.09	94.97	95.02	99.75	3.341	0.281
VGG16	96.60	96.59	96.53	99.92	131.948	14.420
MobileNetV3-Small	97.42	97.33	97.38	99.90	2.425	0.055
MobileNetV2	97.61	97.53	97.55	99.93	3.343	0.293
ResNet18	97.67	97.66	97.64	99.94	11.148	1.694
MobileNetV3-Large	97.73	97.70	97.71	99.93	5.229	0.210
DSGIResNet_AFF	98.30	98.23	98.24	99.97	0.658	0.235

## Data Availability

Not applicable.

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
