# Peer review of "Development of a Lightweight Crop Disease Image Identification Model Based on Attentional Feature Fusion"

_sensors, 2022, doi:10.3390/s22155550_

Round 1
Reviewer 1 Report
The authors of this work focus on reducing the computational demands of well known deep learning methods. Standard convolutional neural networks (CNNs) perform well, but require significant computational resources, which make them sub-optimal for deployment in the field, or on embedded devices. As a use case, the authors choose crop disease image detection, and achieve comparable results to ResNet18, but require significantly less computational power (measured by authors in floating point ops per second, or FLOPS).
The main contribution of this paper is the inverted residual blocks that replace the standard residual blocks with a reduction in FLOPS and the attention feature fusion.
A thorough comparison of different models has been done by the authors, and the model seems to be thoroughly evaluated. The paper is overall well-written and should be accepted for publication, as it provides a contribution to the field.
I would highly recommend that the authors work on figure captions. These are critical pieces of text used by a reader that should explain in enough detail what the above figure represents, without the need to refer back to the main body of the paper. The most appropriate caption is on Figure 11. I suggest the authors redo the captions with at least as much explanatory content as in Figure 11.
Reviewer 2 Report
The paper is clear, and it is relevant for the field of improving disease detection in agriculture. The paper is presented in a well-structured manner, with clearly separated units.
The authors cited most recent publications and given that this is a rapidly developing field, they did an excellent job of that part.
The paper's topic is of interest to scientists involved in applying deep learning in crop disease detection, but I believe it will not have a significant impact in the field. The results and methods presented by the authors do not revolutionize the field, although they indicate the direction in which the models should be improved.
Manuscript's results are reproducible based on the given details, but such papers should also include program code, so interested readers can more easily test the proposed model. Many services allow the sharing of program code, although I believe the authors will do this later.
Figures and tables correctly show the data and are easy to interpret and understand. Considering the type of paper, there is no statistical data processing.
Conclusions are consistent with the evidence and arguments presented.
Finally, I suggest that the authors in part 4.3. (Comparison Experiment of Different Identification Models) if possible, include the MobileNetV3 model and make a comparison with it.
Reviewer 3 Report
Dear Authors
I have Development of a Lightweight Crop Disease Image Identification Model Based on Attentional Feature Fusion send to Sensors. The article is interesting. The subject is very interesting. I have some suggestions to consider before final publication. In the introduction, the Authors did not specify the purpose of the research. What is the utilitarian aspect of research.
Reviewer 4 Report
The paper discusses the crop diseases identification using RESNET 18, although the promising use of RESNET 18 in image classification, this paper suffers critical methodological issues that should be addressed before further consideration. Some of them are listed below
1- The problem statement should be explained well to show the contribution of the paper based on critical review of related works.
2- The related work should be classified based on their properties.
3- Literature survey is not sufficient to present the most updated for further justification of the originality of the manuscript. You should carry out a thorough literature survey of papers published in a range of top energy journals so as to fully appreciate the latest findings and key challenges relating to the topic addressed in your manuscript and to allow you to more clearly present your contribution to the pool of existing knowledge.
4- The related work should be classified based on their properties.
5- The paper should have a novelty. What is the difference between your work and the existing works in literature review?
6- The abstract should be rewritten to show the problem statement, contribution, brief related works, and methodology.
7- Justify fully why a attentional feature fusion is better than using only a deep feature extractor?
8- The results are not sufficient . you should use sensitivity and AUC performance criteria to show the strength of your work.
9- In general, the mathematical background must be carefully checked and discussed.
1- The related work should be classified based on their properties.
1- The conclusion should make a strong case for future directions.
-The references are very few. they should be improved by adding relative articles supporting the research idea.
1- In the text there are errors in English, need to be carefully read and corrected.
Round 2
Reviewer 4 Report
I am satisfied with the revision. I strongly recommended it for publications.
Author Response
We would like to thank you for the time and effort spent in reviewing the manuscript. In response to your suggestions, we have revised the manuscript. We have corrected the English spelling problems in the paper. About the problem of references, we have replaced some of the references to make them more relevant to the study, and the rest of the references are related to the study background and research methods. Please review them in the resubmitted manuscript.